# Neutrophil Extracellular Traps Generation Relates with Early Stage and Vascular Complications in Systemic Sclerosis

**DOI:** 10.3390/jcm9072136

**Published:** 2020-07-07

**Authors:** Kevin Didier, Delphine Giusti, Sebastien Le Jan, Christine Terryn, Celine Muller, Bach Nga Pham, Richard Le Naour, Frank D. Antonicelli, Amelie Servettaz

**Affiliations:** 1EA7509 IRMAIC, University of Reims-Champagne-Ardenne, 51100 Reims, France; dgiusti@chu-reims.fr (D.G.); sebastien.le-jan@univ-reims.fr (S.L.J.); celine.muller@chu-reims.fr (C.M.); bnpham@chu-reims.fr (B.N.P.); richard.le-naour@univ-reims.fr (R.L.N.); frank.antonicelli@univ-reims.fr (F.D.A.); aservettaz@chu-reims.fr (A.S.); 2Department of Internal Medicine, Infectious Diseases, and Clinical Immunology, Reims Teaching Hospitals, Robert Debré Hospital, 51100 Reims, France; 3Laboratory of Immunology, Reims University Hospital, University of Champagne-Ardenne, 51100 Reims, France; 4PICT Platform, University of Reims Champagne-Ardenne, 51100 Reims, France; christine.terryn@univ-reims.fr

**Keywords:** systemic sclerosis, scleroderma, PMN, NETosis, neutrophil extracellular traps, NET

## Abstract

Systemic sclerosis (SSc) is a systemic disease characterized by a great clinical and immunological heterogeneity whose pathophysiology is still being unraveled. Recently, innate immunity has been proposed to participate to the pathogenesis of SSc. In this study, we investigated the release of neutrophil extracellular traps (NETs) according to patient phenotype. Polymorphonuclear neutrophils (PMN) from 34 SSc patients and 26 healthy controls were stimulated by serum from SSc or healthy subject. NETs were visualized using epifluorescence microscope after DNA, myeloperoxidase, and Histone H3 tagging. Area of NETs were quantified using an original macro running in ImageJ^®^ software. PMN from SSc patients were significantly more prone to releasing NETs than control PMN after autologous stimulation. PMN from patients with severe vascular complications (pulmonary arterial hypertension, digital ulcers) produced more NETs than PMN from other SSc patients and their aberrant NET production appeared to be sustained over time. In patients with pulmonary interstitial disease or extensive cutaneous fibrosis, NET production was high at an early stage of the disease before progressively decreasing. Both serum factors and PMN activation status were involved in the enhanced production of NETs in SSc. Consequently, neutrophils and especially NETosis represent new physiopathological and therapeutic fields in SSc.

## 1. Introduction

Systemic sclerosis (SSc) is a rare connective tissue disease characterized by skin and organ fibrosis, vasculopathy, and autoimmunity. There are two clinical subsets of SSc according to the extent of skin fibrosis: limited SSc (lSSc) or diffuse SSc (dSSc) [1]. Besides skin disorders, 20 to 40% of SSc patients develop interstitial lung disease (ILD) that may lead to respiratory insufficiency and death [2,3]. Ischemic digital ulcers, pulmonary arterial hypertension (PAH), and renal microangiopathy (called scleroderma renal crisis) are vascular ravages also observed in some SSc patients. Ischemic digital ulcers represent a chronic and disabling condition, with frequent relapses, whereas scleroderma renal crisis and PAH are life-threatening complications. PAH represents the second cause of death in this setting [2,4]. Survival is heterogeneous in SSc, depending on internal organ involvement [5]. Actually, there is no curative treatment and few drugs have demonstrated efficiency in organ involvement [6,7]. A better comprehension of physiopathology of SSc could improve the medical care of each patient.

The exact pathogenesis of SSc remains unclear, but several defects in the control of TGF-β pathways have been evidenced that may lead to excessive collagen synthesis by fibroblasts and myofibroblasts [8]. Endothelial cells are also altered, leading to early angiogenesis and vasculogenesis abnormalities [8]. Nearly 95% of SSc patients display autoimmunity [3]. Numerous specific autoantibodies (AAb) have been identified in patient sera, among them anti-topoisomerase I AAb (or anti-Scl70 AAb), anti-centromere AAb, and anti-RNA polymerase III AAb [9]. Each autoantibody is mutually exclusive and associated with a specific phenotype [9]. The implication of adaptative immunity is now clearly established, but innate immunity has not been well studied up to now [8,10].

The occurrence of oxidative stress is linked to the pathogenesis of SSc, and polymorphonuclear neutrophils (PMN) are known to generate important quantities of reactive oxygen species (ROS) in the course of this disease [11]. During PMN activation, depending on oxidative stress, cells release DNA filaments covered by histones, enzymes, and other antimicrobial proteins. These filaments, called neutrophil extracellular traps (NETs) are involved in defense against bacteria and their formation leads in most of cases to the death of PMN [12]. This release of NETs, also called NETosis, has also been highlighted in autoimmune diseases such as systemic lupus erythematosus (SLE), rheumatoid arthritis (RA), bullous pemphigoid, and antineutrophil cytoplasmic antibody (ANCA)-associated vasculitis [13,14,15,16]. In these diseases, NETosis seems to be an integral part of pathogenesis, either by promoting tolerance breakdown by facilitating autoantigens exposure, or by directly participating in organ damage with, notably, endothelial dysfunction [17,18]. Recently, it was highlighted that platelet-derived microparticles interacted with PMN, promoting autophagy and the generation of NET byproducts in SSc patients [19].

In this study, we wondered whether NETosis occurring during SSc could be differently modulated according to the phenotype of SSc, reflecting the heterogeneous clinical expression of this disease.

## 2. Patients and Methods

### 2.1. Study Population

Thirty-four patients with SSc were recruited in the Internal Medicine Department at the University Hospital of Reims, France, between January 2018 and February 2020. The study was approved by the ethics committee for the protection of human beings involved in biomedical research (2016/49-ID RCB: 2016-A01517-44). Informed consent to participate was obtained from each patient and control subject included in the study for blood collection and the use of their de-identified medical record data. Patients fulfilled the following criteria for inclusion: age > 18 years, and a diagnosis of SSc according to American College of Rheumatology (ACR)/European League Against Rheumatism (EULAR) criteria [1]. Disease subtype was classified based on LeRoy and Medsger criteria: diffuse cutaneous SSc (dSSc) and limited cutaneous SSc (lSSc). Age- and sex-matched controls (n = 26) were recruited from the French blood Institute.

### 2.2. Data Collection

All variables were entered into a standardized questionnaire fulfilled by the clinician at the time of inclusion. In all patients, the complete medical history was retrospectively reviewed. Physical examination variables, laboratory, and imaging exams were collected for all patients. Disease duration was defined at the time between the first non-Raynaud’s phenomenon symptom and inclusion. Rodnan skin scores were systematically evaluated by the same physician at inclusion. Interstitial lung disease (ILD) was defined as ground-glass opacities and/or interstitial reticular pattern with or without traction bronchiectasis and/or honeycomb cysts on high resolution computed tomography. PAH was diagnosed based on right-heart catheterization if mean pulmonary arterial pressure was ≥25 mmHg and pulmonary capillary wedge pressure was ≤15 mmHg in a patient with either no ILD or ILD with forced vital capacity predicted ≥70% and extent of ILD on HRCT ≤20%. Scleroderma renal crisis was defined as the abrupt onset of severe hypertension and/or decline in renal function, with proteinuria without an alternate etiology.

### 2.3. Antinuclear Antibody Detection

The presence of antinuclear antibodies (ANA) was evaluated by indirect immunofluorescence on HEp-2000 cells-coated slides (Eurobio Ingen, Les Ulis, France) with a positivity threshold set at a titer of 1:100. ANA specificities were then determined by performing both a multiplex fluorescent microsphere immunodetection (FIDIS Connective Profile MX117™, Theradiag^®^, Croissy-Beaubourg, France) and an immunodot assay (Euroline Systemic Sclerosis [Nucleoli] Profile [IgG]; Euroimmun, Luebeck, Germany), allowing separate detection of anti-DNA topoisomerase I, anti-centromere AAb, anti–RNA polymerase III, anti-fibrillarin, anti-NOR90, anti Th/To, anti-PM-Scl, anti-Ku, anti-PDGFR, and anti-Ro-52 antibodies.

### 2.4. Polymorphonuclear Neutrophils (PMN) Isolation and Stimulation

To isolate PMN, heparinized blood was layered on an association of dextran and diatrizoate gradient with a density of 1.113 ± 0.001 (Granulosep: Eurobio, Les Ulis, France) following the manufacturer’s instructions. Cells were then distributed in a 24 well plate (2.10^5^ cells/well) on coverslips coated with L-lysine (*Acros Organics, Belgium*). PMN were then stimulated with serum from SSc patients or from controls (final dilution of 10%) for 4 h at 37 °C and 5% CO_2_ in a final volume of 500 µL. Each condition was performed in duplicate. At the end of incubation, cells were washed and fixed by paraformaldehyde 4% (VWR, Darmstadt, Germany) as described by Brinkmann [20].

### 2.5. Visualization and Quantification of Neutrophil Extracellular Traps (NETs)

Sytox Orange nucleic acid stain (Molecular Probes, Eugene, Oregon, USA) was used to stain extracellular DNA filaments and Hoechst 33342 (Thermo Fisher Scientific, Waltham, Massachusetts, USA) for intra and extracellular DNA labelling, with a dilution of 1/5000 for both in Roswell Park Memorial Institute medium (RPMI) for 15 min. To better characterize NETs, we also performed a double immunostaining with a rabbit polyclonal Ab to Histone H3 (1:100 overnight. Abcam, Cambridge, UK) revealed by a secondary chicken anti-rabbit Ab Alexa Fluor 594 (1:200, one hour, Invitrogen, Carlsbad, USA) and a mouse monoclonal Ab to human myeloperoxydase coupled to a fluorescein isothiocyanate (FITC) fluorochrome (1:100 for one hour, Abcam, Cambridge, UK).

NETs were visualized using an inverted epifluorescence microscope (AxioObserver Z1: Zeiss, Germany; camera Cool Snap HQ2: Roper Scientific, Les Ulis, France) with the software Metamorph (Molecular devices, version 7.7.10, San Jose, CA, USA) at a magnification of 20× with an excitation/emission of 547/570 nm for Sytox Orange and 350/461 nm for Hoechst 33342. On each slide, 16 successive images (on the slide center) of 398 µm by 297 µm were taken for a total area about 2 mm^2^ by coverslips.

The quantification of NETs was done using a home-made macro running in ImageJ software (*ImageJ, U. S. National Institutes of Health, USA*) that counted the number of nucleus stained by Hoechst 33342 and NETs were analyzed using the signal of Sytox Orange. The image was blurred to accentuate edges and to enhance contrast. Then, the image was inverted to obtain an image in black and white. NETs (number and area) were quantified by subtracting the Hoechst 33342 signal of the Sytox Orange signal to remove nucleus in the quantification. This method allowed counting of the total number of PMN nuclei, and DNA filaments and the total area of NETs in the selected zone. Results were analyzed according to the mean number of NETs by 100 PMN and to mean area of NETs by PMN. The method of quantification is described in Figure A1. We decided to investigate the NETosis phenomenon using the area of NETs as proposed by Rebernick et al. [21].

### 2.6. Statistical Analysis

GraphPad Prism 5 (GraphPad, La Jolla, CA, USA) was used for statistical analyses. The variables were not normally distributed, as assessed by the d’Agostino and Pearson omnibus normality test and Shapiro–Wilk normality test, and comparisons between two groups were performed using the Mann–Whitney test. Correlation tests were performed by the Spearman test. To test if disease duration and clinical phenotype were independently associated with the level of NET release, a general linear model was used with the following parameters: disease duration (less than three years versus more than three years) and clinical phenotype (no complication, vascular complications, fibrotic complications, mixed complications). *p* values lower than 0.05 were considered statistically significant.

## 3. Results

### 3.1. Subject Characteristics

The biological and clinical characteristics of the study population are presented in Table 1. As expected, significant differences in Modified Rodnan Score, internal organ involvement, and AAb detection were observed between SSc patients and the controls. No significant difference was observed between the two groups regarding gender and age. Among the SSc patients (n = 34), 26 had lSSc, whereas eight presented with dSSc.

Internal organ involvement was classified as either fibrotic complication (diffuse cutaneous phenotype, and/or ILD) or vascular complication (PAH, and/or scleroderma renal crisis, and/or ischemic digital ulcers), as described in Table 2. Five patients presented both vascular and fibrotic complications. No significant difference was observed concerning follow-up duration between these two SSc groups.

### 3.2. PMN from Systemic Sclerosis (SSc) Patients Generate More NETs Than Those from Controls

Ex vivo NET formation was first investigated by stimulating fresh isolated PMN from patients and controls with sera in autologous conditions. Fluorescence microscopy analysis of stimulated PMN using the DNA-intercaling dyes Sytox Orange and Hoechst 33342 showed that the median area of NETs released by SSc PMN after autologous stimulation was significantly higher than the median area of NETs released by PMN from the controls (8.7 [5.4–14.7] µm^2^/cell versus 1.8 [0–3.5] µm^2^/cell respectively, *p* < 0.0001, Figure 1).

### 3.3. Involvement of Both PMN Activation Status and Serum Factors in the Level of NET Production Is Not Dependent on AAb Specificity in SSc

To further explore the respective contribution of PMN and serum in the excessive NET generation observed in the SSc population, we stimulated PMN from SSc and controls with sera from reciprocal population (Figure 2). PMN from SSc patients generated more NETs than those from the controls when stimulated either with sera from SSc patients (median area of NETs 8.7 [5.4–14.7] µm^2^/cell versus 2.6 [1–5.4] µm^2^/cell, respectively, *p* < 0.0001) or with sera from the controls (median area of NETs 3.5 [1.4–8.2] µm^2^/cell versus 1.8 [0–3.5] µm^2^/cell, respectively, *p* = 0.0090). Moreover, sera from SSc patients induced more NET formation than those from the controls after incubation with the controls’ PMN (median area of NETs 2.6 [1–5.4] µm^2^/cell versus 1.8 [0–3.5] µm^2^/cell, respectively, *p* = 0.0316). Similarly, sera from SSc patients induced more NET formation by SSc PMN than the control sera (*p* = 0.0004).

Given the observed effect of SSc sera on NET generation, we next investigated the potential differences in NET generation according to the specificity of AAb in SSc sera. We showed that the NET area was not significantly different regardless of the AAb detected in sera in autologous conditions of stimulation (Figure 3).

### 3.4. NETosis Is Higher in SSc Patients with Vascular Complications But Is Not Dependent on SSc Cutaneous Phenotype

We next quantified the NET production by SSc PMN according to the disease-related complications in autologous conditions of stimulation. Patients were classified as having either no complication, vascular, fibrotic, or mixed complications (defined by the presence of at least one vascular and one fibrotic complication, Figure 4A). In SSc patients without complication, PMN still generated more NETs than the PMN from the controls (median area of NETs of 7.9 [3.8–12.6] µm^2^/cell versus 1.8 [0–3.5] µm^2^/cell, respectively, *p* = 0.0012). With a median area of 15 [11.5–22.5] µm^2^/cell, PMN from patients with vascular complications released significantly more NETs than those from SSc patients without any complication (median area of 7.9 [3.8–12.6] µm^2^/cell, *p* = 0.0078) or SSc patients displaying fibrotic complications (median area of 7.2 [2.4–11.2] µm^2^/cell, *p* = 0.0079) or SSc patients with mixed complications (median area of 7 [5.6–12] µm^2^/cell, *p* = 0.0186). In contrast, no significant difference was highlighted between patients without complications and patients with fibrotic or mixed complications.

We further investigated NETosis according to the cutaneous phenotypes dSSc and lSSc and did not observe any significant difference between these disease subtypes in autologous conditions of stimulation (*p* = 0.2156, Figure 4B). Furthermore, no correlation was highlighted between NET area and the modified Rodnan skin score, which reflects the spreading of skin fibrosis (Spearman r = 0.1799, *p* = 0.3085).

### 3.5. NETosis Is an Early Process Occurring in SSc That Lasts in SSc Patients with Vascular Complications

A rapid spread of cutaneous and lung fibrosis is classically observed during the first three years of the disease. Consequently, we analyzed NET production according to disease duration. A stimulation of SSc PMN with autologous sera allowed us to demonstrate that NETosis was inversely correlated with the duration of the disease, suggesting that NETosis may occur preferentially early in the disease course (Spearman r = −0.3521, *p* = 0.0412, Figure 5A).

Due to an important heterogeneity highlighted and the impact of the type of complications, we further investigated NET area according to disease duration and the type of complications in autologous conditions of stimulation. As shown in Figure 5B,C, NETosis was inversely correlated with the duration of the disease in patients without complication (Spearman r = −0.5711, *p* = 0.0524) or displaying fibrotic complications (Spearman r = −0.7563, *p* = 0.0214). In contrast, no correlation was highlighted in patients displaying vascular complications (Spearman r = 0.4097, *p* = 0.3135, Figure 5D). The regression model showed a significant interaction between disease duration and clinical phenotype for NET generation (*p* = 0.036 for interaction), confirming that time (disease duration) effect was different according to the phenotype of the disease on NETosis in SSc.

## 4. Discussion

SSc is an autoimmune systemic disease characterized by dysfunctions of endothelial cells, fibroblasts, and immune cells, leading to heterogeneous organ damage that jeopardizes patient survival. The importance of free radicals and ROS in the pathophysiology of SSc, led us to study NETosis in this disease [11]. We here report an excessive NET formation by PMN from SSc patients. Interestingly, aberrant NET generation appears to be closely linked with disease duration and the type of SSc-related complication. Indeed, in patients with recent fibrosis, the high NET formation observed at an early stage of the disease progressively decreased, whereas it remained elevated throughout the entire course of the illness in patients with severe and chronic vascular complications.

Excessive NETosis has already been reported in different noninfectious disorders including autoimmune diseases [22]. In SLE, enhanced spontaneous NETosis was shown, and impaired degradation of NETs was related to an increased risk of developing lupus nephritis [23,24]. Increased NET formation has also been observed in the peripheral blood and synovium of patients with rheumatoid arthritis (RA) and in kidneys and small arterioles of patients with antineutrophil cytoplasmic antibody (ANCA)-associated vasculitis [15,25,26]. Moreover, MPO–DNA complexes and citrullinated H4-histones were previously found in the plasma of SSc patients, suggesting an inappropriate NETosis in this disease [19]. In this study, the authors observed that MPO-DNA complexes were significantly more concentrated in the plasma of patients with an early and active scleroderma pattern compared with those with a late scleroderma pattern in capillaroscopy. In addition, DNA filaments were recently observed in skin biopsies from SSc patients [27]. These possible NET structures were shown to be decorated with CXCL4 and these CXCL4-DNA complexes could participate in immune activation and notably to type-I interferon production in patients with early active SSc. Overall, these two previous works indirectly suggested a role of NETs in some subsets of SSc.

There are many clinical patterns of SSc according to the extent of skin fibrosis, the kind of internal organ involvement, and the type of AAb developed by patients. We here observed that PMN from all SSc patients including those without any organ damage were prone to generate more NETs than PMN from the controls. In addition, in patients with long-lasting vascular damage, the dysregulated NET generation was higher than in other SSc patients and was maintained over time. In contrast, PMN and sera from SSc patients with only fibrotic complications or without any complication appeared less prone to generating NETs.

NET contribution to endothelial dysfunction has been previously reported in other diseases involving vessels. Indeed, in SLE patients, enhanced capacity of PMN to form NETs favored extensive endothelial cell killing and circulating NET level was correlated with arterial thrombosis in antiphospholipid syndrome [23,28,29]. A growing body of evidence also supports a crucial role of PMN and NETs in the development of plaque formation and atherosclerosis [30]. In relation to SSc physiopathology, we can hypothesize that repeated binding of NETs to activated endothelium may induce local production of ROS, inducing endothelial dysfunction, endothelium-dependent vasorelaxation impairment, and endothelial cell apoptosis, all well-documented events early occurring in the pathogenesis of SSc. Noteworthy, the persistent production of NETs highlighted in the group of patients with severe vascular complications could perpetuate vascular remodeling, leading to recurring digital ulcers and incurable PAH [2,4].

These original results may suggest a more prominent role of NETs in endothelial damages than in fibroblast dysfunction. Nevertheless, this conclusion may be hurried. In fact, we observed that PMN from most patients with an early disease (less than three years) were prone to releasing high amounts of NETs including those from patients with only fibrotic complications. Both cutaneous and lung fibrosis develop within the first years of the disease, before stabilizing. Cutaneous fibrosis may even decline after three years [31]. Figure 6 proposes a schematic representation of the relative importance of NET generation we could imagine, according to our data in SSc pathogenesis.

NETosis may be constantly upregulated in patients with severe long-lasting vascular complications whereas the amount of NET production may be particularly aberrant at an early stage of the disease before progressively decreasing along time in patients with a predominant fibrotic pattern. Patients with a mixed phenotype displayed intermediate NET production, but only five were studied, of whom two with renal crisis that had recovered with renin-angiotensin pathway blockade and no other long-lasting vascular complications. We assume that NET generation may play a role in different steps of SSc pathogenesis and that distinct mechanisms may induce and regulate NETosis according to the clinical pattern.

The cascade of events leading to uncontrolled NETosis in SSc remains to be clearly determined. In the present study, we demonstrated that both serum factors and PMN activation status were involved in the enhanced production of NETs in vitro. In serum, the presence of NETosis inducers including AAb, immune complexes, multiple danger-associated molecular patterns (DAMP), and cytokines have been described in various inflammatory diseases. The type of AAb did not seem to influence the level of NET formation in SSc, however we cannot rule out a role of AAbs in SSc-related NETosis since immune complexes were described as NETosis inducers in RA [25]. In addition, ribonucleoprotein-containing immune complexes, which are frequently detected in SSc sera, were shown to induce NETosis in SLE patients and consequently represent interesting serum candidates for SSc NETosis mediators [32]. Conversely, NETs represent a well described source of tolerance break in SLE, RA, and ANCA-associated vasculitis [28,33,34]. Further experiments including sera containing distinct types of AAbs from patients with distinct autoimmune diseases (for example, SSc, RA, and SLE) are needed to further assess the link between autoimmunity and NETosis in SSc. DAMP-expressing microparticles released from activated platelets seem to activate PMN and may also favor NETosis in SSc sera [19]. Regarding SSc PMN, we here found a primed status of these cells leading to an excess of NETosis, as previously highlighted for ROS generation [35]. Several abnormalities among PMN pathways could support aberrant priming that favors NETosis in SSc patients. Indeed, activated ERK, p38, and JNK were present at much higher levels in SSc PMN than in the controls cells, and interestingly, ERK is known to be involved in NET production [36,37]. In addition, IL-8, a crucial cytokine for PMN priming and NETosis, has been found to be increased in SSc sera [12,38]. Regarding treatment able to break uncontrolled NETosis, our study does not allow for any conclusion regarding immunosuppressive or vasodilator drugs, since few patients received such drugs. Identifying NETosis mediators remains the next essential step to further assess therapeutic molecules interfering with NET release and possible toxicity in SSc.

## 5. Conclusions

In conclusion, we demonstrate for the first time excessive and uncontrolled NETosis in SSc PMN, especially at an early stage of the disease during fibrosis progression and in patients with chronic severe vascular complications. Identifying specific NETs mediators according to clinical phenotype is needed to further clarify the role of NETs in the pathogenesis of SSc and its severe complications. If NETosis is confirmed to play a role in some steps of SSc pathogenesis, targeting neutrophils and especially NETosis may open a new therapeutic field in this disease.

## Figures and Tables

**Figure 1 jcm-09-02136-f001:**
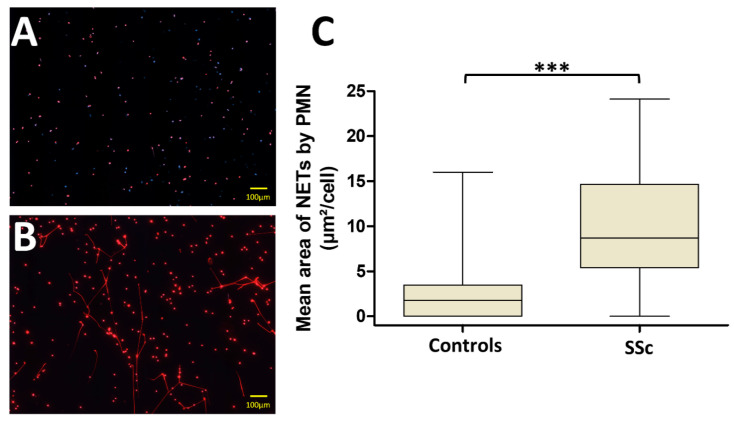
Polymorphonuclear neutrophils (PMN) from systemic sclerosis (SSc) patients generated more neutrophil extracellular traps (NETs) than those from the controls. PMN from SSc patients and controls were stimulated in autologous conditions. (**A**) The most representative pictures (Sytox Orange and Hoechst 33342, 16 consecutives pictures with X20 magnification) of PMN from one control. (**B**) The most representative pictures of PMN from one SSc patient. (**C**) The median areas of NETs were statistically higher in SSc patients than in the controls (n = 34 for the SSc group and n = 26 for the control group). All stimulations were performed in duplicate. The mean area of NETs released by PMN is represented by a “box and whiskers” plot for each condition tested. Boxes show the 25–75th percentiles, lines show the median value, and whiskers show the range of the mean area of NETs released by PMN. *** *p* < 0.001.

**Figure 2 jcm-09-02136-f002:**
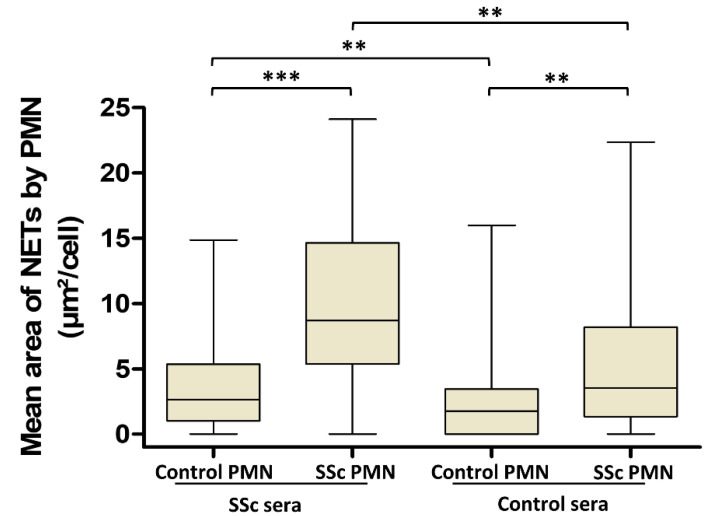
Polymorphonuclear neutrophils (PMN) and serum are both involved in SSc NETosis. To analyze the respective role of cells and sera in NETosis, PMN from SSc patients were stimulated with autologous sera (n = 34) or with sera from the controls (n = 52), and PMN from controls were stimulated with autologous sera (n = 26) or with sera from SSc patients (n = 151). All stimulations were performed in duplicate. The mean area of NETs released by PMN is represented by a “box and whiskers” plot for each condition tested. Boxes show the 25–75th percentiles, lines show the median value, and whiskers show the range of the mean area of NETs released by PMN. *** *p* < 0.001, ** *p* < 0.01.

**Figure 3 jcm-09-02136-f003:**
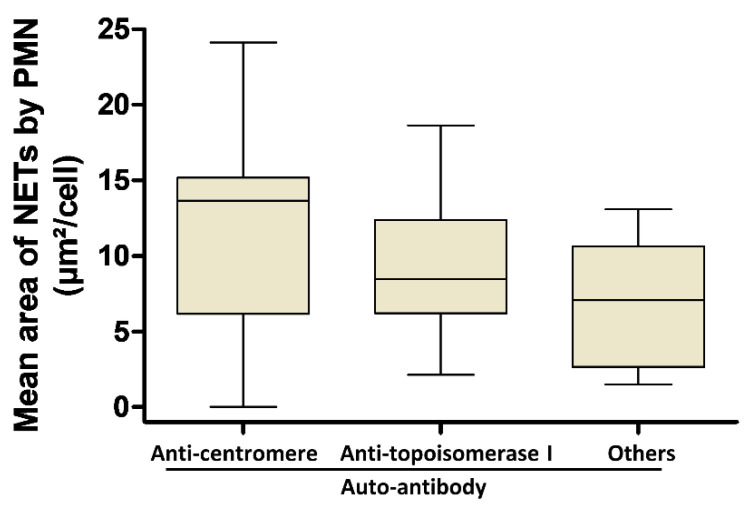
NETosis phenomenon occurring in SSc cannot be explained by the specificity of autoantibody. PMN from SSc patients were stimulated in autologous conditions. Patients were grouped according to the AAb specificity (anti-centromere AAb (n = 17), anti-topoisomerase I AAb (n = 8) or other AAb (n = 9)). All stimulations were performed in duplicate. The mean area of NETs released by PMN is represented by a “box and whiskers” plot for each condition tested. Boxes show the 25–75th percentiles, lines show the median value, and whiskers show the range of the mean area of NETs released by PMN.

**Figure 4 jcm-09-02136-f004:**
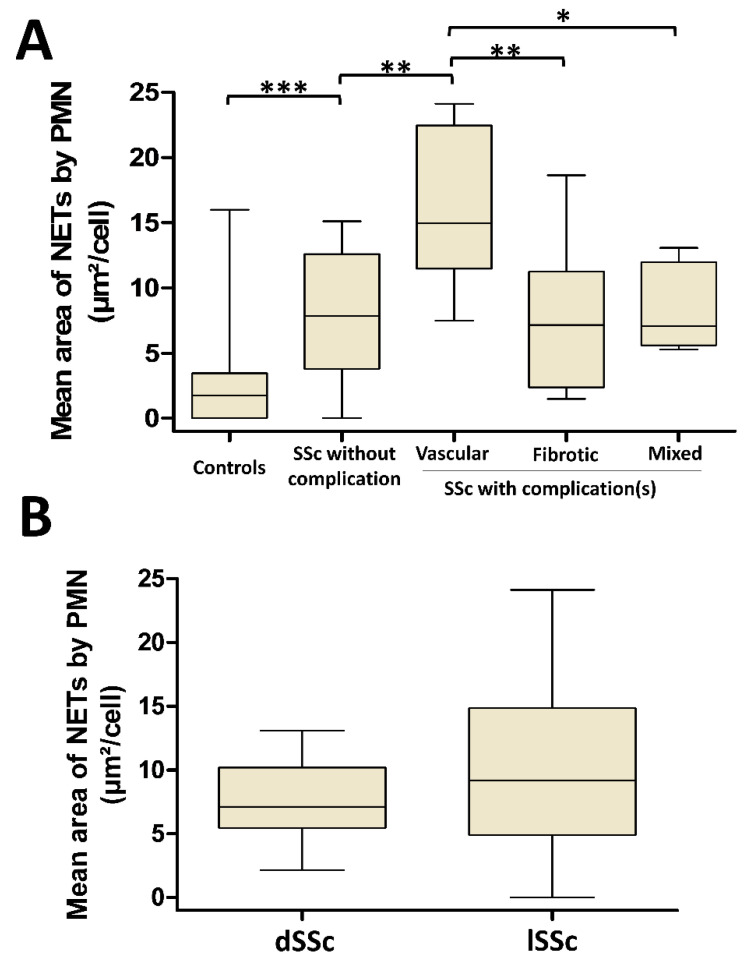
NETosis is increased in SSc patients with vascular complications and is not dependent on the extension of skin fibrosis. (**A**) PMN from SSc patients and controls were stimulated in autologous conditions. Patients were grouped according to their type of complications (vascular, fibrotic, or mixed complications) or the absence of complications. Pulmonary arterial hypertension, digital ulcers, and scleroderma renal crisis were considered as vascular complications, diffuse phenotype and interstitial lung disease were considered as fibrotic complications, and mixed complications were defined as the association of at least one vascular and one fibrotic complication. All conditions (n = 26 for controls, n = 12 for patients without complication, n = 8 for SSc with vascular complications, n = 9 for SSc with fibrotic complications, and n = 5 for SSc with mixed complications) were performed in duplicate. (**B**) Patients were grouped according to their skin phenotype (diffuse SSc (dSSc) or limited SSc (lSSc)) and PMN were stimulated in autologous conditions. All conditions (n = 8 for dSSc, and n = 26 for lSSc) were performed in duplicate. The mean area of NETs released by PMN is represented by a “box and whiskers” plot for each condition tested. Boxes show the 25–75th percentiles, lines show the median value, and whiskers show the range of the mean area of NETs released by PMN. *** *p* < 0.001, ** *p* < 0.01, * *p* < 0.05.

**Figure 5 jcm-09-02136-f005:**
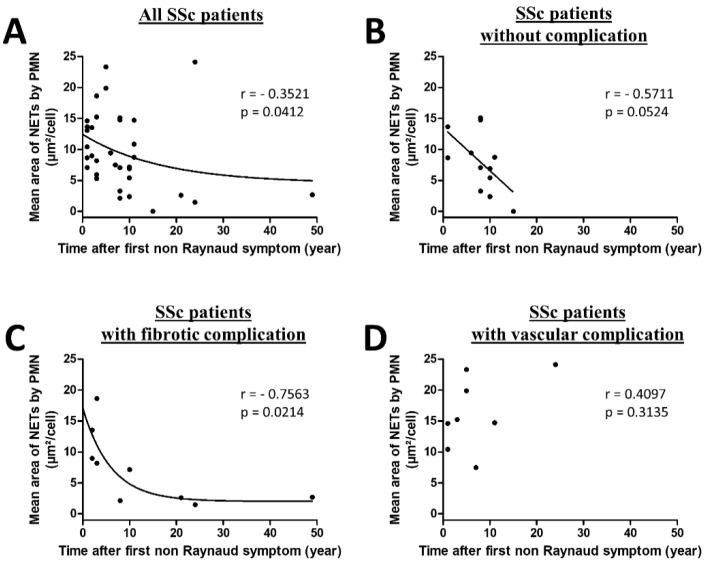
NETosis phenomenon is an early process occurring in SSc patients that persists in patients with vascular complications. (**A**) PMN from patients were stimulated with autologous sera to assess the correlation between mean area of NETs by PMN and time after the first non-Raynaud’s phenomenon symptom. All stimulations (n = 34) were performed in duplicate. (**B**–**D**) Due to a link between the type of complication and quantity of NETosis, results were analyzed according to the type of complication. In each case, PMN from patients was stimulated with autologous sera to assess the correlation between mean area of NETs by PMN and time after the first non-Raynaud’s phenomenon symptom (**B**) without complication (n = 12), (**C**) fibrotic complications (n = 9), and (**D**) vascular complications (n = 8). Each dot represents the mean area of NETs by PMN of two similar experiments. The regression curves concerning all patients (**A**) and patients with fibrotic complications (**C**) were obtained using a one phase exponential decay nonlinear regression with the following equations respectively: Y = (12.45–4.519) × exp(−0.05926X) + 4.519; Y = (17.06–2.028) × exp(−0.1677X) + 2.028. The regression curve concerning patients without complication (B) was obtained using a linear regression with the following equation: Y = (13.50 ± 2.824) + (−0.6936 ± 0.3190)X.

**Figure 6 jcm-09-02136-f006:**
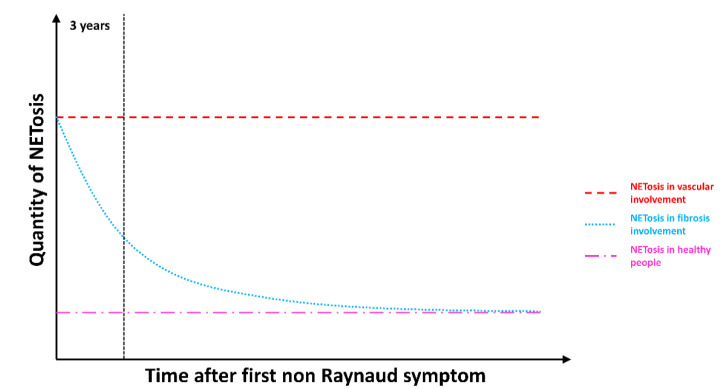
Schematic representation of the quantity of NETs generated by SSc PMN in vitro according to disease duration and the type of complication. The red line represents mean NETosis amount generated by PMN from patients with chronic vascular complications. The blue line represents mean NETosis production by PMN from patients with a predominant fibrotic pattern and no vascular severe complication. The pink line represents mean NETosis production by PMN from healthy subjects.

**Table 1 jcm-09-02136-t001:** Main features of systemic sclerosis (SSc) patients and controls.

Main Features	SSc Patients			Controls
	Whole SSc Population	Diffuse SSc	Limited SSc	
Number of individuals	34	8	26	26
Age (mean, years)	59.5 (±10.9)	55.3 (±11.7)	60.8 (±10.6)	60.5 (±11.2)
Women (%)	29 (85.3)	5 (62.5)	24 (92.3)	19 (73.1)
Disease’s duration (years)	8.6 (±6)	5 (± 3.5)	9.8 (± 6.7)	-
Modified Rodnan score	11.8 (±8.7)	25.4 (± 6.2)	7.6 (± 6)	-
Internal organ involvement				-
Interstitial lung disease	9 (26.5)	3 (37.5)	6 (23.1)	-
Ischemic digital ulcers	8 (23.5)	3 (37.5)	5 (19.2)	-
Pulmonary arterial hypertension	3 (8.8)	0 (0)	3 (11.5)	-
Scleroderma renal crisis	2 (5.9)	2 (25)	0 (0)	-
Auto-antibody				-
Anti-topoisomerase I	8 (23.5)	3 (37.5)	5 (19.2)	-
Anti-centromere	17 (50)	0 (0)	17 (65.4)	-
Anti-RNA polymerase III	2 (5.9)	2 (25)	0 (0)	-
Anti-U1RNP	1 (2.9)	0 (0)	1 (3.9)	-
Anti-U3RNP	2 (5.9)	2 (25)	0 (0)	-
Anti-NOR90	1 (2.9)	1 (12.5)	0 (0)	-
Undetermined	3 (8.8)	0 (0)	3 (11.5)	-
Treatment at the time of blood collection				-
Immunosuppressive drug (including corticosteroids)	2 (5.9)	2 (25)	0 (0)	-
Vasodilator drug	8 (23.5)	1 (12.5)	7 (26.9)	-

**Table 2 jcm-09-02136-t002:** Classification of SSc patients according to the type of complications.

Characteristics	SSc Patients without Complication	SSc Patients with Fibrotic Complications	SSc Patients with Vascular Complications	SSc Patients with both Fibrotic and Vascular Complications
Number of patients	12	9	8	5
Mean of disease duration, year (SD)	8 (±4)	13.6 (±15.6)	7.3 (±7.6)	3.8 (4.2)
Diffuse cutaneous involvement (%)	-	3 (33.3)	-	5 (100)
Interstitial lung disease (%)	-	8 (88.9)	-	1 (20)
Ischemic digital ulcer(s) (%)	-	-	5 (62.5)	3 (60)
Pulmonary arterial hypertension (%)	-	-	3 (37.5)	0 (0)
Scleroderma renal crisis (%)	-	-	0 (0)	2 (40)

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
