# Peer review of "Neutrophil Extracellular Traps Generation Relates with Early Stage and Vascular Complications in Systemic Sclerosis"

_jcm, 2020, doi:10.3390/jcm9072136_

Round 1

Reviewer 1 Report

Here the authors investigated for the first time NETosis in systemic sclerosis (SSc) and found that both SSc polymorphonuclear neutrophils (PMN) as well as SSc sera show or favor enhanced NETosis compared to healthy controls. Disease duration and vascular complications were identified as variables associated with NET formation.

Remarkably, no attempts were made to provide mechanistic explanations for these observations.

  1. As stressed by the authors, NETosis has been described in several systemic autoimmune disorders. While their data convincingly provide evidence for enhanced in vitro NETosis in SSc, in their manuscript there is a lack  quantitative data to understand how SSc compares to SLE or other diseases. Would the authors consider to test and compare SLE and SSc sera on healthy PMN for providing such an estimation?
  2. Could the authors provide histology data for the existence of NETosis in SSc tissues? If not, could they provide literature evidence for such an occurrence?
  3. Based on correlative evidence, the data here presented tend to exclude a role for autoantibody specificities in SSc-related NETosis. However formal testing for the role of autoantibodies or immunecomplexes is lacking. Could the authors provide data on the effect of purified SSc Ig vs sera depleted of Ig on NETosis? If not the discussion should stress further this limitation of the study.
  4. In Figures 1, 2, 3, and 4 the data are reported as mean and confidence interval (SEM?). However, if I understand correctly, the data are deemed as non normally distributed. Thus Medians and 25-75% box-plots may be more appropriate for the representation of the data.
  5. Since in some experiments the author tested SSc PMN combined with SSc sera and in others SSc PMN with healthy sera, to avoid any confusion it would be appropriate to stress which combination they use for each specific set of experiments.
  6. Relatively few SSc individuals were under glucocorticoid or immunosuppressant treatments. Nonetheless, treatment may affect PMN behaviour and propensitiy to undergo NETosis. Thus, it will be useful to address the question whether treatment affected PMN NETosis in their patients.
  7. In Fig 5 it is unclear which correlation method was used and how the regression curves were calculated. Please, specify.
  8. The authors provide data indicating that NETosis is particularly important early in the disease course and in individuals with vascular complications, with those presenting mixed fibrosis and vasculopathy in between. Could the authors perform a multivariable analysis to ascertain whether their graphic representation (Fig 6) has a stronger statistical background? Is timing or organ involvement the most important factor?
  9. Could the authors discuss the possible mechanisms explaining why the mixed clinical phenotype (fibrosis+vasculopathy) fells in between the two pure phenotypes?

Author Response

Review 1 : Comments and suggestions for Authors

Here the authors investigated for the first time NETosis in systemic sclerosis (SSc) and found that both SSc polymorphonuclear neutrophils (PMN) as well as SSc sera show or favor enhanced NETosis compared to healthy controls. Disease duration and vascular complications were identified as variables associated with NET formation. Remarkably, no attempts were made to provide mechanistic explanations for these observations.

1. As stressed by the authors, NETosis has been described in several systemic autoimmune disorders. While their data convincingly provide evidence for enhanced in vitro NETosis in SSc, in their manuscript there is a lack quantitative data to understand how SSc compares to SLE or other diseases. Would the authors consider to test and compare SLE and SSc sera on healthy PMN for providing such an estimation?

We agree with the reviewer and we plan to compare our results obtained in SSc to SLE. As we observed that both serum factors and neutrophil activation status determined the level of NETosis, we envisage to test and compare SLE and SSc sera on healthy PMN, and healthy sera on SLE and SSc PMN (Clinical trial identification number NCT03374618). Several works have been published on NETosis in SLE, and consequently a body of proofs shows uncontrolled NET production together with impaired NET degradation in SLE patients. Moreover, NET release has been shown to be associated with disease activity and kidney involvement in SLE (Frangou E, et al. Autoimmun Rev. 2019;18(8):751-760). Nevertheless, it is difficult to compare our results with those obtained in SLE by other groups because many conditions differ among the protocols used to trigger and measure NETs. A single study in our knowledge has simultaneously quantified MPO-DNA complexes by enzyme-linked immunosorbent assay in plasma from SSc (N=57) and SLE patients (N=6) (Maugeri N, et al. Sci Transl Med. 2018;10(451), reference 19 in the manuscript). MPO-DNA complexes concentrations (expressed in arbitrary units) seem similar in the two groups although this point was not clearly discussed by the authors. Importantly, the authors suggested that the factors involved in NETosis were different between the two groups as they showed that platelet-derived microparticles from SSc patients trigger neutrophil activation and maybe NET production. By contrast microparticles retrieved from the blood of patients with SLE were weaker inducers of neutrophil activation. No detail was given concerning disease activity for SLE and complications for SLE and SSc patients.

As others, we hypothesize that NETosis triggers, mechanisms, level and NET composition are depending on the pathophysiologic context of each disease. Consequently comparison of NETosis in different groups of patients with distinct and well-phenotype diseases is one of the future crucial steps.

This point of discussion has been added to the manuscript as following (page 12 lines 352-355).

2. Could the authors provide histology data for the existence of NETosis in SSc tissues? If not, could they provide literature evidence for such an occurrence?

To gain further insights into the role of NETs in tissue injury, we totally agree with the reviewer that the observation of NETs in damaged organs is a strong argument. To our knowledge, no study has been performed on SSc tissues, but different observations report NET structures in biopsy samples from patients with inflammatoty diseases. By example, NETs were found in kidney biopsies of patients with proliferative lupus nephritis and in skin biopsies from patients with active discoid lupus (Frangou E, et al. Ann Rheum Dis 2019;78:238–248). Skin biopsies obtained either from non-inflamed skin of the same patients with SLE or from healthy subjects did not demonstrate the presence of NETs. NETs have also been observed in thrombi from patients with myocardial infarction, with sepsis or with thrombotic microangiopathies (De Boer OJ, et al. Thromb Haemost 2013;109:290–7 ;  Frangou E, et al. Autoimmun Rev. 2019;18(8):751-760). NETs were also identified in vasculitic small arterioles from nasal and nerve biopsy samples and in pulmonary capillaries of ANCA-positive patients (Takeuchi H, et al. Clin Rheumatol. 2017;36(4):913–7 ; Matsuda Y, et al. Pathol Int. 2016;66(8):466-71). Lesional skin biopsy specimen from patients with pemphigoid bullous also contained NETs in the papillary dermis at the edge of the dermal-epidermal separation (Giusti D, et al. Front. Immunol. 2019 10:701).

According to these data of the literature, we could hypothesize that such neutrophil traps may be found in involved skin area of SSc patients at an early stage of the disease. Alveolar fluid and lung interstitium may also contain NETs in patients with active lung fibrosis. Pulmonary arterioles and capillaries also represent interesting area to investigate especially in patients with pulmonary hypertension. Nevertheless, all these investigations need tissue biopsies, that are rarely performed in SSc, contrary to lupus or vasculitis.

3. Based on correlative evidence, the data here presented tend to exclude a role for autoantibody specificities in SSc-related NETosis. However formal testing for the role of autoantibodies or immunecomplexes is lacking. Could the authors provide data on the effect of purified SSc Ig vs sera depleted of Ig on NETosis? If not the discussion should stress further this limitation of the study.

We totally agree with the reviewer that further testing the role of autoantibodies or immune complexes is needed to assess their ability to enhance or not NETosis in SSc as this was described in RA, SLE and ANCA-associated vasculitis (Khandpur R, et al. Sci Transl Med. 2013 27;5(178):178ra40 ; Carmona-Rivera C, et al. Ann Rheum Dis. 2015;74(7):1417–24 ; Panda R, et al. Front Immunol. 2017;8:439 ;  Pratesi F, et al. Ann Rheum Dis. 2014;73(7):1414–22). We did not observe any significant difference in NET release regardless of the AAb detected in sera of SSc patients. Nevertheless, we agree that we can absolutely not rule out a role of some AAbs in SSc-related NETosis. Further specific experiments are planned to assess this point by using either purified IgG or depleted sera.

This point has been added to the discussion paragraph (page 12; lines 346-352).

4. In Figures 1, 2, 3, and 4 the data are reported as mean and confidence interval (SEM?). However, if I understand correctly, the data are deemed as non-normally distributed. Thus Medians and 25-75% box-plots may be more appropriate for the representation of the data.

We agree with the reviewer. Medians and 25-75% box plots are more appropriate for the representation of our non-normally distributed data. Consequently we change this important point in all figures and in the text.

5. Since in some experiments the author tested SSc PMN combined with SSc sera and in others SSc PMN with healthy sera, to avoid any confusion it would be appropriate to stress which combination they use for each specific set of experiments.

We are in agreement with the reviewer and clarify this point in the results part of the manuscript and in the legend of the figure 5 (page 7, lines 201-202 ; page 8, lines 213 and 225-226  ; page 10, lines 255-256 and 269-273).

6. Relatively few SSc individuals were under glucocorticoid or immunosuppressant treatments. Nonetheless, treatment may affect PMN behaviour and propensitiy to undergo NETosis. Thus, it will be useful to address the question whether treatment affected PMN NETosis in their patients.

We agree with the reviewer that the possible effect of drugs on NETosis should be discussed.

Among the 34 SSc patients, only two received immunosuppressive drug at the time of blood collection: one was taken 10 mg a day of prednisolone, whereas the second one received mycophenolate mofetil. The effect of steroids on NETosis has been poorly investigated to date and results are conflicting (Vargas et al. Respiratory Research. 2017. 18:207; Lapponi M.J. et al; J Pharmacol Exp Ther 2013. 345:430–437; Luan S.X., Eur Rev Med Pharmacol Sci 2017;21(4):855-860). We can suppose that the effect of steroids on NETosis is depending on the mechanisms involved in the process and may be variable according to the medical context. No data has been published concerning mycophenolate mofetil to our knowledge. Some data regarding other types of immunosuppressive drugs suggest that calcineurin antagonists could suppress or modulate NETosis (Gupta AK, et al.  PLoS ONE 2014 9(5): e97088)

Eight patients received vasodilatator drug for PAH and/or digital ulcers. Most of them received endothelin receptor antagonist. Three had a combination of endothelin receptor antagonist and phosphodiesterase-5 (PDE5) inhibitor. Frangou et al recently observed that endothelin-1 may be one of the mediators of NETosis in PMN from active SLE patients and that bosentan, an endothelin receptor antagonist, reduced NET production (Frangou E, et al. Ann Rheum Dis 2019;78:238–248). In our hands, seven out of eight patients in the group with severe vascular complications received endothelin receptor antagonist, and after autologous stimulation, PMN from this group of patients produced higher level of NETs than PMN from the other groups. We could hypothesized that NET production would be even higher in the absence of treatment with endothelin receptor antagonist. Nevertheless, our study was not designated to address the effect of bosentan on NETosis and we totally agree that the question needs a specific study given the promising results obtained in SLE in vitro.

This point has been added to the discussion paragraph (page 13; lines 362-365).

7. In Fig 5 it is unclear which correlation method was used and how the regression curves were calculated. Please, specify.

For the data presented in figure 5, correlation tests were performed by the Spearman test. Concerning regression curves few works have investigated the level of NET production according to continuous data (time) and none included regression curves. Moreover many analyses on NETosis did not use reproducible methods of quantification. For all these reasons we had to select a model to apply for the first time in this purpose. We select the best model whose curve fits the best the data in each case. In practice we observed that NETosis was different according to each clinical phenotype, so we separated the data according to the clinical phenotype. We knew X (time) precisely, and we assumed that the variability in Y (area of NETs) was random, and consistent all the way along the curve. Moreover, all observations were independent, and each data point contributed independent information. In the case of patients without any complication, linear regression was used given the distribution of the data, using the following equation: Y = (13.50 ± 2.824) + (-0.6936 ± 0.3190) * X. For data concerning all patients and patients with fibrotic complications, we choose nonlinear regression that better fits to data distribution and that also may reflect the natural course of fibrosis in the group of patients with fibrosis (one phase exponential decay nonlinear regression, corresponding to the following equations respectively: Y = (12.45-4.519) * exp(-0.05926 * X) + 4.519 ; Y = (17.06-2.028) * exp(-0.1677 * X) + 2.028).

These elements have been added to the Figure 5 legend (pages 10 and 11; lines 276-281).

8. The authors provide data indicating that NETosis is particularly important early in the disease course and in individuals with vascular complications, with those presenting mixed fibrosis and vasculopathy in between. Could the authors perform a multivariable analysis to ascertain whether their graphic representation (Fig 6) has a stronger statistical background? Is timing or organ involvement the most important factor?

We thank the reviewer for this suggestion and performed an additional multivariate analysis to test if the duration of the disease and the clinical phenotype were independently associated with the level of NET release. We used a general linear model with the following parameters: disease duration (less than three years versus more than three years), clinical phenotype (no complication, vascular complications, fibrotic complications, mixed complications).

As depicted in the figure below, the regression model demonstrated a significant interaction between disease duration and clinical phenotype for NET generation (p = 0.036 for interaction). Thus, this analysis reinforces the point we suggested in the discussion and in the figure 6 as it confirms that the effect of time (disease duration) on NETosis is different according to the phenotype of the disease. NETosis appears constantly up-regulated in patients with vascular complications whereas the amount of NET production is high at an early stage of the disease before decreasing along the time in patients with a fibrotic pattern.

This point has been added to the method paragraph (page 4, lines 140-144) and to the results paragraph (page 10; lines 260-262).

9. Could the authors discuss the possible mechanisms explaining why the mixed clinical phenotype (fibrosis+vasculopathy) fells in between the two pure phenotypes?

 We thank the reviewer for his remark. We have no definitive explanation for this phenomenon, but we can propose the following elements to supply the discussion: the group only contains five patients, consequently it is difficult to draw final conclusion. Moreover, among these patients, two were classified with vascular complications since they had presented with renal crisis. None patients with a pure vascular phenotype had renal crisis (see Table 1). Contrary to PAH and recurrent digital ulcers, renal crisis is probably a more acute complication and consequently deregulated NETosis could decrease after renal microangiopathy recovery with angiotensin-converting enzyme inhibitors. Last but not least, we can assume that NETosis triggers and mechanisms may differ according to each group of SSc patients and consequently regarding only at the levels of NET production may be insufficient to fully clarify the pathogenesis of each type of complication. We plan to further explore NETosis stimuli and NET composition for each group. 

 This point has been added to the discussion paragraph (page 12; lines 337-341).

Reviewer 2 Report

The authors present a very well-designed study on Netosis in SSc patients. The methods are described appropriately and the presentation of the results is fluent. The most intriguing result is related to SSc patients with vascular complications, since they appear to have higher levels of Nets compared to both controls and other SSc subgroups. It would be interesting however to analyze the association between first vascular complication according to each patients more than first non-Raynaud symptom, which is widely accepted but it doesn't fit in the flow of the study. The relevance of vascular complication according to NEts should be better discussed in the discussion as well, as in the actual state is too redundant. 

Author Response

The authors present a very well-designed study on Netosis in SSc patients. The methods are described appropriately and the presentation of the results is fluent. The most intriguing result is related to SSc patients with vascular complications, since they appear to have higher levels of Nets compared to both controls and other SSc subgroups. It would be interesting however to analyze the association between first vascular complication according to each patients more than first non-Raynaud symptom, which is widely accepted but it doesn't fit in the flow of the study.

We thank the reviewer for his suggestion and perform three additional analyses.

In a first one we analyzed the association between the beginning of Raynaud’s phenomenon and NETosis level for the 34 patients since all presented with this vascular symptom. No significant correlation was found (Spearman r = -0.2517, p = 0.1511).

In a second analysis, we analyzed the association between the first non-Raynaud’s severe vascular symptom and NET production in the group of patients with at least one severe vascular complication and no fibrotic complication (N=8). No significant correlation was found (Spearman r = 0.4097, p = 0.3268).

Finally, we tested the association between the first non-Raynaud’s severe vascular symptom and NET production in the group of patients with at least one severe vascular complication with or without fibrotic complication (N=13, including 8 patients with a vascular phenotype and 5 patients with a mixed phenotype). No significant correlation was found (Spearman r = 0.1824, p = 0.5510).

Consequently, these supplemental analyses support the absence of correlation between any vascular symptoms duration and the level of NET production in SSc.

The relevance of vascular complication according to NEts should be better discussed in the discussion as well, as in the actual state is too redundant. 

We agree with the reviewer and modified the part dedicated to the relationship between vascular complications and NETosis in the discussion.

This part has been modified in the discussion paragraph (page 11; lines 302-319).

Reviewer 3 Report

This is an interesting and well performed study that demonstrates excessive and uncontrolled NETosis in SSc PMN, specially at an early stage of the disease during fibrosis progression and in patients with chronic severe vascular complications.

Although the results are clear and significant, it would be really interesting to identify the mechanisms underlying the development of NETosis in these patient groups. This would help to identify both the induction mediators and, subsequently, the consequences of the increase in NETosis or its maintenance over time.

To this end, it would be necessary to evaluate the pro-inflammatory / pro-fibrotic profile present in the sera of all the groups evaluated with different clinical profiles (i.e. by using a multiplex assay). This aspect would really add quality and stronger conclusions to the results of study.

It would also be interesting to evaluate the influence of different drugs on NETosis, since it ha been widely demonstrated in other pathologies that several drugs (i.e. corticosteroids, biological drugs) reduce the production of NETosis. This aspect could perhaps explain the reduction observed in more advanced stages of the disease.

Author Response

This is an interesting and well performed study that demonstrates excessive and uncontrolled NETosis in SSc PMN, specially at an early stage of the disease during fibrosis progression and in patients with chronic severe vascular complications.

Although the results are clear and significant, it would be really interesting to identify the mechanisms underlying the development of NETosis in these patient groups. This would help to identify both the induction mediators and, subsequently, the consequences of the increase in NETosis or its maintenance over time. To this end, it would be necessary to evaluate the pro-inflammatory / pro-fibrotic profile present in the sera of all the groups evaluated with different clinical profiles (i.e. by using a multiplex assay). This aspect would really add quality and stronger conclusions to the results of study.

We thank the reviewer for his comments and suggestions concerning the mechanisms underlying the development of aberrant NETosis in SSc, especially in patients with vascular complications. We totally agree that our data represent a first step in the field and that the mechanisms involved in this phenomenon need to be further unraveled. We demonstrated that both serum factors and the activation status of neutrophils determined the level of NET production in SSc. We will rapidly test the role of different cytokines, of auto-antibodies, of advances oxidation protein products and of regulators of autophagy (notably endothelin) since some of them have been shown to be involved in the regulation of NETosis and are known to be upregulated in SSc sera (Servettaz A et al. Ann Rheum Dis 2007;66:1202–1209. ; Frangou E, et al. Ann Rheum Dis 2019;78:238–248). As suggested by the reviewer, we think that it is necessary to evaluate a large panel of candidates since they may differ according to each clinical profile, and disease duration and consequently, we plan to combine multiplex assays and specific approaches.

It would also be interesting to evaluate the influence of different drugs on NETosis, since it has been widely demonstrated in other pathologies that several drugs (i.e. corticosteroids, biological drugs) reduce the production of NETosis. This aspect could perhaps explain the reduction observed in more advanced stages of the disease.

We agree with the reviewer that the possible effect of drugs on NETosis should be discussed.

Among the 34 SSc patients, only two received immunosuppressive drug at the time of blood collection: one was taken 10 mg a day of prednisolone, whereas the second one received mycophenolate mofetil. Given the few numbers of patients receiving immunosuppressive treatment we hypothesize that this point had few impacts on our results but additional experiments addressing the effect of immunosuppressive drugs on NETosis in SSc are needed. Regarding the literature, the effect of steroids on NETosis has been poorly investigated to date and results are conflicting (Vargas et al. Respiratory Research. 2017. 18:207; Lapponi M.J. et al; J Pharmacol Exp Ther 2013. 345:430–437; Luan S.X., Eur Rev Med Pharmacol Sci 2017;21(4):855-860). We can suppose that the effect of steroids on NETosis is depending on the mechanisms involved in the process and may be variable according to the medical context. No data has been published concerning mycophenolate mofetil to our knowledge. Some data regarding other types of immunosuppressive drugs suggest that calcineurin antagonists could suppress or modulate NETosis (Gupta AK, et al.  PLoS ONE 2014 9(5): e97088).

Eight patients received vasodilatator drug for PAH and/or digital ulcers. Most of them received endothelin receptor antagonist. Three had a combination of endothelin receptor antagonist and phosphodiesterase-5 (PDE5) inhibitor. Frangou et al recently observed that endothelin-1 may be one of the mediators of NETosis in PMN from active SLE patients and that bosentan, an endothelin receptor antagonist, reduced NET production (Frangou E, et al. Ann Rheum Dis 2019;78:238–248). In our hands, seven out of eight patients in the group with severe vascular complications received endothelin receptor antagonist, and after autologous stimulation, PMN from this group of patients produced higher level of NETs than PMN from the other groups. We could hypothesize that NET production would be even higher in the absence of treatment with endothelin receptor antagonist including patients in more advanced stages of the disease. Nevertheless, our study was not designated to address the effect of bosentan on NETosis and we totally agree that the question needs a specific study given the promising results obtained in SLE in vitro.

This point has been added to the discussion paragraph (page 13; lines 362-365).

Round 2

Reviewer 1 Report

In the revised manuscript the authors addressed most if not all my queries. As points of detail, I have two remarks.

  1. At variance with the statements made in the answer at point 2, evidence for the presence of NETs in SSc tissues by has been published and could be quoted since it reinforces their message: NATURE COMMUNICATIONS | (2019) 10:1731 | https://doi.org/10.1038/s41467-019-09683-z
  2. Since box-plots have been adopted to represent their findings, the sentence: "Each dot represents the mean observed after 2 similar experiments" present in the legends of figures 1, 2, 3, and 4 should be eliminated and replaced by the meaning of the box-plots and whiskers

Author Response

In the revised manuscript the authors addressed most if not all my queries. As points of detail, I have two remarks.

  1. At variance with the statements made in the answer at point 2, evidence for the presence of NETs in SSc tissues by has been published and could be quoted since it reinforces their message: NATURE COMMUNICATIONS | (2019) 10:1731 | https://doi.org/10.1038/s41467-019-09683-z

We agree with the reviewer and resume the results obtained by this group in the discussion, as they observed NETs in skin biopsies from SSc patients, and brought evidence for a role of plasma and tissue CXCL4-DNA complexes in immune activation in SSc (lines 305-309).

  1. Since box-plots have been adopted to represent their findings, the sentence: "Each dot represents the mean observed after 2 similar experiments" present in the legends of figures 1, 2, 3, and 4 should be eliminated and replaced by the meaning of the box-plots and whiskers.

We apologize and correct this mistake in the legend of the figures 1, 2, 3 and 4.

Reviewer 3 Report

The authors have appropriately answered to all my concerns

Author Response

The authors have appropriately answered to all my concerns

We would like to thank the Reviewer 3 for all the advices given that will improve our study